# Longitudinal Analysis of Intrinsic Capacity and Other Risk Factors in Aging: FREVO Study Protocol

**DOI:** 10.3390/healthcare13090993

**Published:** 2025-04-25

**Authors:** Rodrigo Cappato de Araújo, Letícia Bojikian Calixtre, Wildja de Lima Gomes, Juliana Daniele de Araújo Silva, Diógenes Candido Mendes Maranhão, Fernando Damasceno de Albuquerque Angelo, Gabriel Lucas Leite da Silva Santos, Késia Moreira Sampaio Amaral, Ruth Lahis da Silva Gonçalves, Julia Gomes de Alencar, Michele L. Callisaya, Francis Trombini-Souza, Ana Carolina Rodarti Pitangui

**Affiliations:** 1Graduate Program of Rehabilitation and Functional Performance, University of Pernambuco (UPE), Petrolina 56328-900, PE, Brazil; leticia.calixtre@upe.br (L.B.C.); wildja.gomes@upe.br (W.d.L.G.); juliana.daniele@upe.br (J.D.d.A.S.); diogenesmendes.maranhao@upe.br (D.C.M.M.); fernandodamasceno.angelo@upe.br (F.D.d.A.A.); gluc.91@gmail.com (G.L.L.d.S.S.); kesia.sampaio@upe.br (K.M.S.A.); ruth.goncalves@upe.br (R.L.d.S.G.); julia.gomesalencar@upe.br (J.G.d.A.); francis.trombini@upe.br (F.T.-S.); carolina.pitangui@upe.br (A.C.R.P.); 2Department of Physical Therapy, University of Pernambuco (UPE), Petrolina 56328-900, PE, Brazil; 3National Centre for Healthy Ageing, Melbourne, VIC 3199, Australia; michele.callisaya@monash.edu; 4Peninsula Clinical School, Central Clinical School, Monash University, Frankston, VIC 3199, Australia; 5Menzies Institute for Medical Research, University of Tasmania, Hobart, TAS 7005, Australia

**Keywords:** intrinsic capacity, aging, longitudinal study, gerontology, risk factors, public health

## Abstract

**Background/Objectives:** Population aging presents important global and socio-economic challenges, especially in developing countries such as Brazil, where aging is projected to accelerate in the next years. This manuscript presents the protocol of the FREVO (risk factors in aging) study, a six-year longitudinal study that aims to assess intrinsic capacity and its interaction with other risk factors. Moreover, this study aims to evaluate the combination of these factors and their correlation with major adverse health outcomes among community-dwelling older adults in Petrolina, Brazil. **Methods:** This six-year prospective cohort study will recruit 496 participants aged 60 or older. Annual in-person assessments using validated tools will measure intrinsic capacity, personal information, lifestyle, and chronic conditions. Negative outcomes (falls, hospitalizations, dementia, and death) will be recorded biannually by phone. Statistical analyses will employ latent profile analysis to identify risk phenotypes and Cox regression models for time-to-event analyses. **Results:** This study will attempt to identify phenotypes and modifiable risk factors by using the WHO’s intrinsic capacity framework in a low-resource Brazilian context for the assessment and promotion of healthy aging. **Conclusions:** Our findings will address important gaps that can contribute to a localized understanding of aging, aligning global frameworks with regional realities to promote independence, functionality, and quality of life for older adults.

## 1. Introduction

Population aging is a global phenomenon that has become increasingly significant in recent decades, generating socioeconomic impacts and challenges for healthcare systems worldwide [1]. This process has occurred gradually in developed nations, whereas in emerging economies, it is progressing rapidly [1]. The World Health Organization (WHO) estimates that approximately 70% of older adults will reside in developing countries. Currently, Brazil has approximately 32 million older adults, representing just over 15% of the total population. This number is expected to more than double by 2050, when projections estimate that the older adult population will exceed 65 million, accounting for nearly 30% of the country’s population [2]. This demographic shift is occurring in the context of persistent social inequalities and limited healthcare coverage in many regions. For instance, in the semi-arid Northeast region, where this study was conducted, the proportion of older adults is growing rapidly, with fewer resources available to support healthy aging. These regional disparities highlight the urgent need for context-specific approaches to aging [2].

The aging process is accompanied by declines in mobility, balance, and sensory and psychological functions. During this process, significant deterioration in one or more of these domains, combined with comorbidities, lifestyle habits, and socio-environmental factors, can pose additional risks for developing geriatric syndromes, hospitalizations, and death [3,4,5]. Given this scenario, the World Health Organization (WHO) proposed the Integrated Care for Older People (ICOPE) strategy, which promotes healthy aging by maintaining functional ability through early identification and management of intrinsic capacity decline. Intrinsic capacity is defined as the composite of an individual’s physical and mental abilities, encompassing five key domains: cognition, mobility, vitality, psychological well-being, and sensory function [5,6,7]. Unlike traditional disease-based models, intrinsic capacity focuses on positive attributes associated with successful aging, supporting a proactive and person-centered approach to care [8].

Each domain contributes to preserving autonomy and ensuring quality of life in older adults. The decline in cognitive functions, ranging from mild cognitive impairment (MCI) to various dementia conditions, can exert a significant bidirectional influence on other domains of intrinsic capacity. These conditions frequently expose older people to an increased risk of negative health outcomes [9,10]. Environmental and clinical factors may further affect cognition, and modifying these factors has the potential to enhance both intrinsic capacity and functional ability [11].

Physical function is considered a key component in multidimensional geriatric assessment, as it exhibits a linear correlation with the risk of various negative health outcomes. Evidence demonstrates that simple measures, such as gait speed, can effectively predict functional and cognitive decline and even provide an estimate of life expectancy in older adults [12].

The assessment of psychological well-being focuses on mood disorders, particularly depressive symptoms, which may present subclinically and may be associated with a decline in functional capacity. Depressive symptoms may serve as an independent risk factor or interact with other conditions, contributing to declines in intrinsic capacity and increasing the risk of adverse health outcomes [13,14,15].

The domain relating to sensory capacity considers deficits in vision and hearing, which impair the global health and functional performance of older adults. Impairments in these capabilities compromise the ability to engage in social interactions and perform daily activities. Decreased or lost visual acuity is associated with depressive symptoms and an increased risk of falls [16]. Although more prevalent than visual impairments, hearing loss is less associated with decreased physical performance. However, they are correlated with functional impairment due to difficulties in communication and the consequent social isolation. Furthermore, decreased auditory function is a risk factor associated with dementia [17,18,19].

The vitality domain assesses modifications in metabolism and energy expenditure using anthropometric markers such as body mass index and nutritional status. Changes in nutritional status or metabolic function during aging constitute risk factors for frailty, sarcopenia, and osteoporosis [20].

These domains are interrelated and may interact synergistically with others and contribute to the risk of negative health outcomes such as frailty, dementia, and falls. In addition, other factors, such as chronic pain, cardiovascular and metabolic diseases, physical inactivity [21], and urinary incontinence [22], have also been identified as relevant to the functional decline of older adults.

In Brazil, the implementation of this evaluation methodology could facilitate the inclusion of alternative, complementary, and low-cost biomarkers in population evaluation and monitoring processes within the context of primary care. Moreover, most information regarding the relationship between personal, behavioral, and environmental aspects and health outcomes in the aging process is derived from research conducted in developed countries. In this context, it is essential to consider the role of personal, social, and environmental factors in the aging process, particularly within the realities of a developing country like Brazil. This becomes particularly relevant in regions characterized by lower levels of human development, where health conditions and social determinants exert a significant influence on the aging trajectory.

Therefore, the primary objective of this study is to evaluate intrinsic capacity, identify associated risk factors, and estimate the prevalence of variables. This study will investigate the relationship between intrinsic capacity and key adverse health outcomes among older adults in Petrolina, Pernambuco, over six years.

This study hypothesizes that by analyzing the interplay among the five domains of intrinsic capacity and other relevant risk factors, it will be possible to more accurately identify distinct subgroups or phenotypes of older adults at a significantly increased risk of experiencing adverse health outcomes associated with aging. This improved identification of high-risk individuals will facilitate the implementation of targeted interventions aimed at preventing or delaying the progression of functional decline. Consequently, this approach has the potential to optimize health, enhance functional performance, and promote independence among older adults, ultimately contributing to an improved quality of life and healthier aging.

## 2. Materials and Methods

### 2.1. Study Design

This is a five-year prospective longitudinal study, conducted at the University of Pernambuco (UPE), titled the FREVO study. FREVO is an acronym for Fatores de Risco no Envelhecimento, the original name of this Portuguese study, which translates to Risk Factors in Aging.

### 2.2. Population and Sample

This study will be conducted with a sample of individuals aged 60 years or older, of both sexes, who use primary healthcare services and reside in Petrolina, Pernambuco (Brazil). The sample size calculation was based on the estimated prevalence of falls (25.1%) in national studies [23], considering a sampling error of 5%, a confidence level of 95%, a design effect of 1.5, and a loss to follow-up rate of 15%. Thus, the estimated minimum sample size was 496 participants. The sample size was calculated based on the prevalence of falls (25.1%), the most frequent adverse outcome in community-dwelling older adults. We acknowledge that this may not provide sufficient statistical power for less frequent outcomes, such as dementia or mortality. These outcomes will be analyzed in an exploratory manner, and power calculations will be conducted to inform the interpretation of the results. The sample distribution by sex and age was projected based on population data and is detailed in Table 1.

### 2.3. Eligibility Criteria

Older adults of both sexes living in the community, aged 60 years or older, who use primary healthcare services and reside in Petrolina, Pernambuco, will be recruited from 2024 to 2028 to participate in this study. To be included, they should present themselves in the laboratory for baseline data collection, with or without the assistance of third parties, and sign the informed consent form. Older adults with a history of serious cardiovascular disease, cognitive impairment, or neuromuscular diseases will not be included in the sample.

Older adults who do not respond to phone calls during the study period or are unable to attend the laboratory once a year for follow-up assessments, even with assistance, will be excluded from the sample.

### 2.4. Ethical Aspects

This study was approved by the Research Ethics Committee of the Amaury de Medeiros Integrated Health University Center (CISAM/UPE), under CAAE number 69901423.9.0000.5191. Informed consent will be obtained from all participants before enrollment. The collected data will be securely stored in REDCap (Research Electronic Data Capture).

### 2.5. Study Procedures

Eligible community-dwelling older adults will be invited to participate in a screening assessment at their designated healthcare centers. At these sites, researchers will verify eligibility criteria, explain the study, and present the informed consent document for signature. Part of the baseline assessment measures will also be collected. To ensure the feasibility and quality of data collection, particularly given the comprehensive nature of the assessments, procedures will be organized to minimize participant fatigue. When necessary, data collection will be divided into two sessions held on separate days. On the first day, priority will be given to collecting sociodemographic information and performing the intrinsic capacity assessment, including the cognitive, mobility, vitality, sensory, and psychological domains. On the second day, complementary assessments will be performed, including measures of health status, comorbidity screening, pain, urinary incontinence, nutritional status, and functional performance tests. This two-step approach enhances participant comfort and engagement, promoting more accurate and comprehensive data collection.

Participants will be invited to attend annual laboratory assessments for four consecutive years, employing the same instruments used at baseline. Follow-up phone calls will be conducted every six months to record the occurrence of adverse outcomes, including mortality, falls, hospital admissions, and confirmed diagnoses of dementia. To minimize loss to follow-up, the study will adopt a multi-pronged retention strategy, including regular updating of participant and caregiver contact information, periodic reminder calls, collaboration with community health agents, flexible scheduling with the option for home visits or split assessment sessions when needed, non-financial incentives such as health feedback and small appreciation tokens, and systematic tracking of participant status over time to monitor and address potential attrition. Subsequently, a detailed description of the variables and data collection tools will be provided (Table 2).

### 2.6. Variable and Data Collection Tools

#### 2.6.1. Demographic and Health-Related Characteristics

A standardized questionnaire will be administered to collect data from the participants. This questionnaire will gather sociodemographic information, including self-declared gender, skin color, education level, monthly income, housing conditions, and family support network. Additionally, it will assess the participants’ health history, encompassing current or previous smoking and alcohol consumption, previous clinical diagnoses, presence of comorbidities, current medications, history of falls within the past year, and prior hospitalizations.

#### 2.6.2. Activities of Daily Living

The Brazilian OARS Multidimensional Functional Assessment Questionnaire (BOMFAQ) [13] will assess self-reported difficulty in performing 15 daily activities. These activities encompass eight activities of daily living (ADLs): getting in/out of bed, eating, combing hair, walking on the ground, showering, dressing, maintaining continence, and ascending stairs. Additionally, seven instrumental activities of daily living (IADLs) are included: adhering to medication schedules, walking short distances outside the home, shopping, preparing meals, cutting toenails, driving, and cleaning the house. For each activity, the participants will indicate whether they experience “no difficulty” or “some difficulty”. A higher total score on the BOMFAQ signifies greater functional impairment.

The Barthel Index will be employed to assess basic activities of daily living (ADLs), specifically evaluating the ability to independently perform tasks such as eating, bathing, dressing, maintaining personal hygiene, bowel and bladder continence, toilet use, bed-to-chair transfers, walking, and stair climbing. The Barthel Index yields a score ranging from 0 to 100, with higher scores indicating greater independence.

The Lawton Instrumental Activities of Daily Living Scale assesses eight instrumental ADLs, including travel, shopping, meal preparation, housekeeping, medication management, and financial management. The scale generates a summary score ranging from 0 (low function) to 8 (high function), classifying participants as totally dependent, partially dependent, or independent.

#### 2.6.3. Level of Physical Activity

The Brazilian short version of the International Physical Activity Questionnaire (IPAQ) will be utilized to classify the participants’ physical activity levels as high, moderate, or low [24]. This classification will be based on the combined duration of walking and moderate to vigorous physical activity reported across the assessment period.

#### 2.6.4. Sarcopenia and Frailty

Sarcopenia will be assessed using the SARC-F [25], a questionnaire that explores five dimensions: muscle strength, need for assistance with ambulation, rising from a chair, climbing stairs, and history of falls. Each item is scored on a 3-point Likert scale (0 to 2), resulting in a total score ranging from 0 to 10. According to the literature, a score of 4 or higher is considered predictive of sarcopenia.

The Clinical Frailty Scale will be used to assess frailty, assigning participants a score from 1 to 9 based on their overall health status. A score of 1 indicates “very robust” individuals who are highly active and energetic, while a score of 9 signifies “terminal” individuals with a life expectancy of less than 6 months. Intermediate scores represent varying degrees of frailty, ranging from “robust” individuals with controlled diseases to “severely frail” individuals who are completely dependent on others.

#### 2.6.5. Intrinsic Capacity—Cognition

Global cognitive function will be assessed using the Brazilian version of the Montreal Cognitive Assessment (MoCA) [26]. The MoCA evaluates cognitive domains, including short-term memory (delayed recall), visuospatial skills (cube drawing, clock drawing), executive functions (trail-making test part A, phonemic fluency, verbal abstraction), attention, concentration, and working memory (cancellation, digit span), language (naming, sentence repetition), and orientation (time, space). The MoCA yields a total score ranging from 0 to 30, with higher scores indicating better cognitive function. An additional point will be added to the raw score for participants with less than 12 years of education. Following Brazilian validation studies, we will apply education-adjusted cutoff points for the MoCA to minimize misclassification in participants with lower educational attainment [26,27].

#### 2.6.6. Intrinsic Capacity—Vitality

Nutritional status will be assessed using the Mini-Nutritional Assessment (MNA). The initial screening includes six items evaluating food intake, weight loss, mobility, the presence of significant emotional factors impacting diet, neuropsychological problems, and body mass index. This screening yields a maximum score of 14 points. Scores of 12–14 indicate normal nutritional status, 8–11 suggest risk of malnutrition, and 0–7 indicate malnutrition [28].

Following the initial screening, a further 12 questions are assessed, including measurements of calf and arm circumference. The final MNA score ranges from 0 to 30. A score of 24–30 indicates normal nutritional status, 17–23.5 suggests a risk of malnutrition, and a score below 17 indicates malnutrition [28].

#### 2.6.7. Intrinsic Capacity—Psychological Aspects

Depressive symptoms will be assessed using the Geriatric Depression Scale (GDS-15) [29]. A GDS-15 score of 5 or higher will be considered indicative of depressive symptoms. While the GDS-15 does not provide a formal diagnosis, it is a validated screening tool for depressive symptoms in older adults and has demonstrated reliability in the Brazilian population.

#### 2.6.8. Intrinsic Capacity—Mobility

The Short Physical Performance Battery (SPPB) [30] will be utilized to assess functional performance and mobility. The SPPB combines assessments of gait speed, static balance, and lower limb strength. The balance test assesses the participant’s ability to maintain static balance for 10 s in three positions: feet together, semi-tandem, and tandem. The final SPPB score ranges from 0 to 12, with higher scores indicating better performance. Participants are classified as having poor (0–3 points), low (4–6 points), moderate (7–9 points), or good (10–12 points) functional performance [30].

The Timed Up and Go (TUG) test will be performed in both a single-task (regular) and dual-task condition [31]. The regular TUG assesses the time required for the participant to rise from a chair, walk three meters to a designated line, turn around, return to the chair, and sit down [31]. Participants are instructed to walk as quickly and safely as possible.

To assess gait speed, participants will walk 6 m at their usual pace. The route will be marked with cones in an open space: 2 m for acceleration, 6 m for timed walking, and 2 m for deceleration. The stopwatch will start at the second mark (start of the 6-m mark) and pause at the third mark (start of deceleration), recording the time to walk only the central 6 m [30].

The dual-task mobility tests incorporate a concurrent cognitive task. While performing the regular TUG and 6 m walk test, participants are instructed to count backward from a randomly selected number between 20 and 100, or name animals or fruits [32]. Before the dual-task mobility tests, a 10 s cognitive task is administered while the participant stands still to assess cognitive performance independent of walking [33]. During the dual-task mobility tests, participants are instructed to divide their attention between walking and cognitive tasks. Dual task cost (DTC) will be calculated to quantify the impact of the cognitive task on gait performance using the following formula [33]:DTC (%) = [(Regular TUG time − Dual-task TUG time)/Regular TUG time] × 100

Handgrip strength will be assessed using a JAMAR handgrip dynamometer, as recommended by the American Society of Hand Therapists. Three measurements will be taken, and the highest value will be recorded for analysis [34].

#### 2.6.9. Intrinsic Capacity—Sensorial Function

Visual acuity will be assessed using the Snellen Chart [35], a validated tool suitable for use in resource-limited settings and easily administered by non-specialized healthcare professionals.

Hearing acuity will be assessed using the whisper test. The examiner, positioned posteriorly (from 60 cm) and laterally to the participant’s ear, will whisper four unrelated standardized words. The right ear will be tested first, with the left ear occluded by gentle pressure on the left tragus. Subsequently, the left ear will be tested using the same procedure. Failure to correctly repeat two or more words will be indicative of hearing loss [36].

#### 2.6.10. Sleep Quality

Sleep quality will be assessed using the Brazilian version of the Pittsburgh Sleep Quality Index (PSQI) [37]. The PSQI is a self-reported questionnaire consisting of 19 items and 5 additional items completed by a bed partner, which are used for clinical information only. The 19 self-reported items are grouped into seven components: subjective sleep quality, sleep latency, sleep duration, habitual sleep efficiency, sleep disturbances, use of sleep medication, and daytime dysfunction. Each item is scored on a scale of 0 to 3.

The scores of these components are summed to generate a global PSQI score, which ranges from 0 to 21. Higher scores indicate poorer sleep quality. A global PSQI score greater than 5 suggests clinically significant sleep disturbances, defined as severe difficulties in at least two components or moderate impairment in more than three components [37].

#### 2.6.11. Urinary Incontinence

The International Consultation on Incontinence Questionnaire—Short Form (ICIQ-UI SF) [38] will be used to assess the presence and severity of urinary incontinence (UI) as well as its impact on quality of life. The ICIQ-UI SF is a widely used self-administered questionnaire that evaluates UI symptoms, including frequency, severity, and impact on daily life, in both males and females. Scores range from 0 to 21, with higher scores indicating greater severity of UI.

To differentiate between stress and urge incontinence, the Questionnaire for Urinary Incontinence Diagnosis (QUID) [39] will be administered. The QUID is a self-administered questionnaire comprising six items with a response scale ranging from “never” (0) to “every time” (5). The first three items assess stress incontinence, while the last three assess urge incontinence.

The QUID provides domain-specific scores ranging from 0 to 15. A score of 4 or higher in the stress domain indicates stress incontinence, while a score of 6 or higher in the urge domain suggests urge incontinence [39].

#### 2.6.12. Pain

The Brief Pain Inventory (BPI) [40] will be used to assess the presence, intensity, and interference of pain. The presence will be determined based on the first item of the BPI. Pain intensity will be evaluated using items 3 to 6 of the instrument. Pain interference will be assessed using item 9, which represents the average interference of pain across seven domains: general activity, mood, walking ability, work, social relationships, sleep, and enjoyment of life. These domains assess the degree to which pain interfered with these aspects of daily life over the past week.

#### 2.6.13. Life Space Assessment

The Life-Space Assessment (LSA) [41], a self-reported questionnaire translated and cross-culturally adapted for the Brazilian population [41,42], will be used to assess participants’ mobility profiles within their living environments. Five life-space levels are considered: (1) bedroom, (2) external areas of the residence, (3) neighborhood, (4) city of residence, and (5) other cities. For each level, the questionnaire assesses both the frequency of movement (less than once per week, one to three times per week, four to six times per week, or daily) and the degree of independence (without equipment or assistance, with assistive equipment, or with personal assistance). The LSA score ranges from 0 to 120, calculated by summing the scores across five distinct life-space levels. Higher scores indicate greater mobility within the living environment. The validity, reliability, and interpretability of the LSA, particularly concerning content validity criteria, have been demonstrated in community-dwelling older adults in Brazil [43].

The psychometric properties of the Brazilian version of the LSA, including validity, reliability, and interpretability, have been previously established [41]. Specifically, internal consistency, as measured by Cronbach’s alpha, was 0.92. Reliability, assessed by intraclass correlation coefficient (ICC), was 0.97 (95% CI: 0.95–0.98), with a standard measurement error of 4.12 [42].

#### 2.6.14. Blood Pressure

Vital signs will be monitored during the assessments to ensure participant safety. Brachial blood pressure will be measured using an electronic and digital monitor (HEM-742, Omron Healthcare, Kyoto, Japan). Individuals will remain in the supine position for ten minutes, and three consecutive measurements will be taken, one minute apart, on both arms using a cuff of a size appropriate to the circumference of the arm. The average of the last two measurements will be used for analysis, following the recommendations of the Brazilian Society of Cardiology.

#### 2.6.15. Anthropometric Measurements

Body mass and height will be measured using a standardized scale and a stadiometer, respectively. The body mass index (BMI) will be calculated by dividing body mass in kilograms by the square of height in meters.

Anthropometric measurements will be taken, including abdominal circumference at the midpoint between the last rib and the iliac crest, hip circumference at the level of the iliac crests, neck circumference above the cricoid cartilage, calf circumference at the maximum point of the dominant calf with the knee flexed at a 90-degree angle, and arm circumference at the midpoint between the acromion and olecranon processes, with the arm extended along the body and the palm facing the thigh. All measurements will be performed while the participant is standing upright, except for calf circumference, which will be measured with the knee flexed. Each measurement will be taken three times, and the average will be used for analysis.

#### 2.6.16. The Sitting and Rising Test

The sitting and rising test (TSLC) is a method to assess functional capacity in non-hospitalized adults, predicting mortality in people aged 51 to 80 years. The assessment involves counting the number of supports (hands and/or knees) used to sit down on and rise from the floor, with separate scores assigned to each movement. Each action is scored on a scale of up to 5 points, with a deduction of 0.5 points for any noticeable loss of balance. If the individual needs extra assistance or more than four supports, the score is zero. The best result of the two attempts for each of the acts will be adopted as representative of the individual [44].

#### 2.6.17. Concerns About Falls

The Falls Efficacy Scale International (FES-I) will be used to assess the degree of concern about falling during activities of daily living, social interactions, and tasks involving postural control. It consists of 16 items scored on an ordinal scale (1 = not at all worried; 2 = little worried; 3 = very worried; and 4 = extremely worried), with a total score ranging from 16 to 64 [45].

### 2.7. Outcome Measures

The occurrence of falls and/or near-falls will be monitored every six months through telephone interviews with participants. In addition to recording the number of fall events, detailed information will be collected regarding the date, time, location, and circumstances of each fall.

The incidence of hospitalizations and deaths will be ascertained through biannual telephone interviews with participants or their designated family members. Additionally, the reasons for hospitalization and the clinical diagnosis received will be inquired about. For cases of death, the cause of death will be recorded.

Possible cases of dementia will be identified through several methods: confirmation of a clinical diagnosis of dementia by family members or caregivers, documentation of an ICD-10 code for dementia in the participant’s medical record, and documentation of prescriptions for cholinesterase inhibitors. Additionally, participants with a MoCA score ≤ 15 and impairments in activities of daily living will be referred to neurologists or geriatricians for further evaluation to confirm a diagnosis of dementia.

## 3. Statistical Analysis

Baseline variables will be presented as means and standard deviations for continuous variables and absolute and relative frequencies for categorical variables. Comparisons between groups will be performed using the chi-square test for categorical variables and Student’s *t*-test for independent samples for continuous variables. To analyze univariate and multivariate associations, prevalence ratios of the factors under investigation concerning the loss of intrinsic capacity will be estimated using a modified Poisson regression with a robust estimator.

Latent profile analysis (LPA) is a versatile analytical approach that enables the identification of patterns in data and the inference of unobserved sources of heterogeneity within a population. It is a person-centered analytical strategy that focuses on similarities and differences between people rather than relationships between variables. This strategy aims to group participants with similar characteristics into distinct profiles based on intercorrelated variables [46].

LPA uses response patterns in dichotomous variables to estimate two parameters: latent class probabilities and conditional probabilities. Latent class probabilities reflect the prevalence of each class, while conditional probabilities indicate the rate of each variable, given membership in a latent class. These estimates allow for calculating the individual probability of class membership based on symptom patterns and the modal class [46].

In this way, the aim is to obtain dichotomous classifications concerning the decline of cognitive, motor, and sensory functions, as well as other personal, behavioral, and social variables. The final LPA model will be determined based on a consensus of multiple fit criteria—BIC value, entropy, LRT, and adjusted LRT. After conducting the LPA and identifying the most efficient number of classes, each participant will be allocated to the class with the highest likelihood of membership.

Cox proportional hazards regression models will be applied to estimate the hazard ratio (HR) and the associated 95% confidence interval (CI) for the occurrence of adverse health outcomes. The proportional hazards assumption will be verified using Schoenfeld residuals. Cox regression models will include age, sex, education, multimorbidity, and baseline functional status as covariates. Additional exploratory models will examine interactions with latent profiles of intrinsic capacity. The analyses may be stratified and adjusted by other variables when the theoretical and statistical criteria are verified. All analyses will be conducted using the JAMOVI 2.3.21 software, with a significance level of 5% (*p* < 0.05).

Due to the longitudinal nature of the study and the advanced age of the participants, some degree of loss to follow-up and missing data is anticipated. To mitigate the potential bias introduced by missing values, we will adopt a multiple imputation strategy using chained equations (MICE) under the assumption that data are missing at random (MAR). This method is widely recommended in longitudinal cohort studies and allows for uncertainty in the imputation process by generating multiple plausible datasets and pooling the results across them [47].

Multiple imputation remains a robust and accessible technique in cohort study settings, particularly when the degree of missingness is moderate and the relationships among variables are complex. Sensitivity analyses will be conducted to compare the results obtained with and without imputation. Additionally, we will explore machine learning methods such as K-nearest neighbors (KNN) and Random Forest imputation [47].

At the end of the follow-up period, the overall loss-to-follow-up rate will be calculated and reported. We will assess whether differential attrition occurred by comparing baseline characteristics between participants lost to follow-up and those retained in the cohort. Additionally, we will examine the potential impact of attrition on study outcomes.

## 4. Considerations

This study aims to provide a comprehensive analysis of the sociodemographic, physical, functional, cognitive, and psychological characteristics of community-dwelling older adults in Petrolina, Pernambuco. It also explores the concept of intrinsic capacity within a sociocultural and environmental context that presents specific regional characteristics.

The research is being conducted in a semi-arid area marked by significant social inequality. This region has been largely overlooked in national studies. By examining aging in this underrepresented setting, the study contributes to filling a key knowledge gap related to the impact of environmental and socioeconomic challenges on older adults.

With rigorous methodology and the use of validated instruments, this study proposes not only to estimate the prevalence of intrinsic capacity decline among older adults, but also to investigate the associated risk factors, aiming to understand the specific challenges faced by this population. Additionally, the study aims to identify factors associated with adverse events, such as dementia, falls, hospitalizations, and mortality, providing essential elements for the development of evidence-based public policies and interventions.

By exploring the application of the intrinsic capacity concept, our study will test its applicability within a specific Brazilian context. This effort enables the adaptation and validation of global strategies to local reality, contributing to the broader application of this conceptual approach.

The findings of this study have the potential to inform the planning of health policies and geriatric care in the region and in similar contexts, enabling more effective interventions in primary care and support for healthy aging. By identifying modifiable risk factors, our study offers data to inform strategies for the prevention and early management of conditions associated with declines in functional and cognitive capacities, strengthening efforts to protect and restore the health of older adults.

Despite its strengths, this study is subject to certain limitations. The use of self-reported data introduces the potential for recall bias. Moreover, the longitudinal design may be affected by participant attrition over time. Although efforts will be made to minimize losses to follow-up, some degree of bias may remain.

It is also important to note that the sample comprises only community-dwelling older adults living in the urban area of Petrolina, a city located in the semi-arid region of northeastern Brazil. As a result, the findings may not be generalizable to other Brazilian regions with distinct sociocultural and infrastructural characteristics.

Future studies should aim to include participants from rural settings and cities with diverse socioeconomic profiles to enhance the external validity of the findings and support the development of more inclusive aging policies across different Brazilian contexts.

We acknowledge that the exclusion of older adults with severe cardiovascular, neuromuscular, or cognitive conditions, although methodologically justified to ensure participant retention and data quality, may limit the generalizability of our findings. These individuals represent a high-risk group whose needs are often underrepresented in aging studies. Future analyses or complementary studies may be warranted to explore strategies for including such populations in longitudinal aging research. However, it would require substantial adjustments to recruitment procedures, ethical protocols, measurement instruments, and data collection logistics.

Although the FREVO study incorporates several individual-level and contextual variables, we acknowledge limitations in capturing broader systemic and dynamic factors, such as changes in the healthcare system, environmental stressors, or political and economic shifts, that may affect the aging process over time. When possible, we plan to incorporate contextual indicators (e.g., local healthcare access, climatic variables) into future waves of data collection or linked analyses to enhance our understanding of the aging context in resource-limited settings.

In summary, our study not only expands the scientific knowledge regarding aging in vulnerable contexts but also provides practical contributions and supporting strategies to promote the well-being and quality of life of older adults. Furthermore, the findings of this study have the potential to inform the planning of evidence-based health policies and geriatric care strategies in Brazil, particularly in underserved and low-resource regions. By aligning the WHO’s global frameworks with the realities of the Brazilian healthcare system, the FREVO study may contribute to more equitable, feasible, and impactful approaches to promoting healthy aging at the community level. Finally, the results may serve as a foundation for guiding primary care interventions, informing clinical decision-making, and supporting training and policy efforts across the country.

## Figures and Tables

**Table 1 healthcare-13-00993-t001:** Distribution of the sample according to sex and age ranges.

Age Range	Population	%	Required Sample
Men 60–69 years old	9555	25.3	125
Women 60–69 years old	11,754	31.1	154
Men 70–79 years old	4879	12.9	64
Women 70–79 years old	6363	16.8	84
Men 80–89 years old	1752	4.6	23
Women 80–89 years old	2602	6.9	34
Men ≥ 90 years old	291	0.7	4
Women ≥ 90 years old	584	1.5	8
TOTAL	37,780	100	496

**Table 2 healthcare-13-00993-t002:** Timeline for data collection of all study variables.

Data	Baseline	6 Months	12 Months	18 Months	24 Months	30 Months	36 Months	42 Months	48 Months
Sociodemographic	X		X		X		X		X
Health-Related Characteristics	X		X		X		X		X
Activities of Daily Living	X		X		X		X		X
Physical Activity	X		X		X		X		X
Sarcopenia and Frailty	X		X		X		X		X
IC-Cognition	X		X		X		X		X
IC-Vitality	X		X		X		X		X
IC-Psychological Aspects	X		X		X		X		X
IC-Mobility	X		X		X		X		X
IC-Sensorial Function	X		X		X		X		X
Sleep Quality	X		X		X		X		X
Urinary Incontinence	X		X		X		X		X
Pain	X		X		X		X		X
Life Space Assessment	X		X		X		X		X
Anthropometrics	X		X		X		X		X
Blood Pressure	X		X		X		X		X
The Sitting and Rising Test	X		X		X		X		X
Concerns about Falls	X		X		X		X		X
Falls	X	X	X	X	X	X	X	X	X
Hospital Admissions	X	X	X	X	X	X	X	X	X
Deaths	X	X	X	X	X	X	X	X	X
Dementia Diagnoses	X	X	X	X	X	X	X	X	X

“X” indicates the time points at which each variable was collected.

## Data Availability

Data is contained within the article.

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
