# Peer review of "Longitudinal Analysis of Intrinsic Capacity and Other Risk Factors in Aging: FREVO Study Protocol"

_healthcare, 2025, doi:10.3390/healthcare13090993_

Round 1

Reviewer 1 Report

Comments and Suggestions for Authors

Peer Review Report

Manuscript Title: Longitudinal Analysis of Intrinsic Capacity and Other Risk Factors in Aging: FREVO Study Protocol

General Assessment:
The manuscript presents a well-structured study protocol for a longitudinal analysis of intrinsic capacity and risk factors in aging. The research topic is highly relevant, especially in the context of aging populations in low-resource settings. The study's potential contributions to public health policies and gerontology are commendable. However, certain methodological and conceptual aspects require further clarification and refinement to strengthen the manuscript’s scientific robustness.

Major Concerns:

  • The introduction provides a good contextualization of the study but contains redundancies regarding the concept of intrinsic capacity. A more concise presentation would improve readability.

  • Sample Size Justification: The sample size calculation is based on the prevalence of falls (25.1%), but since multiple outcomes (hospitalization, mortality, dementia) are being analyzed, further justification regarding the statistical power for each outcome is necessary.

  • Follow-up Strategies: The protocol includes annual in-person assessments and biannual phone calls. However, a detailed plan to minimize loss to follow-up is necessary to ensure the study’s internal validity.

  • Feasibility of Data Collection: The manuscript describes a comprehensive assessment protocol covering multiple domains of intrinsic capacity, health status, and functional performance. However, it is unclear how the researchers will manage such an extensive evaluation in a single session. Will the assessments be conducted on separate days to reduce participant fatigue? Clarifying the logistics of data collection would enhance the study's feasibility and replicability.

  • The study employs validated instruments, which is a strength. However, some limitations should be discussed:

    • MoCA for Cognitive Assessment: The cutoff of <26 may lead to false positives in populations with low education. The adjustment for education levels should be explicitly addressed.

  • The Cox proportional hazards regression model is appropriate for time-to-event analysis, but it is unclear which variables will be included for adjustments.

  • The manuscript does not discuss how missing data will be handled, which is critical in longitudinal studies. A missing data management strategy should be specified.

  • The manuscript should avoid the use of the term elderly, as it is considered ageist. Instead, terms like "older adults" or "aging population" should be used to ensure inclusivity and avoid unintended biases.

  • With these improvements, the study protocol will offer a more solid contribution to the field of gerontology and public health.

Author Response

Dear Reviewer,

Thank you very much for your thoughtful and constructive feedback. We are pleased that you found the study protocol to be well structured and relevant to current public health and gerontological challenges, particularly in low-resource settings. Your insights are greatly appreciated and have helped us enhance the clarity, feasibility, and scientific rigor of the manuscript. Below, we address each of your points:

  • The introduction provides a good contextualization of the study but contains redundancies regarding the concept of intrinsic capacity. A more concise presentation would improve readability.

We appreciate your observation regarding the repetitions related to the concept of intrinsic capacity. We have revised the Introduction to streamline the content, reducing redundancies while maintaining a clear rationale for the study.

  • Sample Size Justification: The sample size calculation is based on the prevalence of falls (25.1%), but since multiple outcomes (hospitalization, mortality, dementia) are being analyzed, further justification regarding the statistical power for each outcome is necessary.

The sample size calculation was based on the most prevalent outcome—falls—among community-dwelling older adults, which offers a conservative estimate. However, we have now clarified this rationale and discussed the limitations regarding statistical power for secondary outcomes (hospitalization, mortality, dementia). These outcomes will be explored primarily in descriptive and exploratory analyses, and we acknowledge that future studies with larger samples will be needed for more robust modeling.

  • Follow-up Strategies: The protocol includes annual in-person assessments and biannual phone calls. However, a detailed plan to minimize loss to follow-up is necessary to ensure the study’s internal validity.

We agree that retention is critical for internal validity in longitudinal studies. We have now added a more detailed description of our strategies to minimize loss to follow-up, including regular contact through caregivers, community health workers, reminder calls, and small engagement incentives.

  • Feasibility of Data Collection: The manuscript describes a comprehensive assessment protocol covering multiple domains of intrinsic capacity, health status, and functional performance. However, it is unclear how the researchers will manage such an extensive evaluation in a single session. Will the assessments be conducted on separate days to reduce participant fatigue? Clarifying the logistics of data collection would enhance the study's feasibility and replicability.

Thank you for raising this important point. We have clarified in the Methods section that data collection will be organized in two sessions, scheduled on separate days when needed, to reduce participant fatigue. This adjustment enhances the feasibility and quality of the data collected.

MoCA Cutoff Adjustment

We agree that the standard MoCA cutoff may not be appropriate for individuals with lower education. As such, we have included an explanation that education-adjusted cutoffs will be applied, as recommended by the instrument's Brazilian validation studies.

Cox Regression Covariates

We have now specified that the Cox regression models will adjust for age, sex, education, multimorbidity, and baseline functional status, and that additional exploratory models will assess interactions with intrinsic capacity profiles.

Missing Data Handling

Thank you for highlighting this essential issue. We have revised the Statistical Analysis section to describe our missing data management strategy, which includes the use of multiple imputation by chained equations (MICE), with sensitivity analyses to assess the impact of missingness on outcomes.

Use of Age-Inclusive Language

We appreciate your attention to inclusive and non-ageist language. We have corrected the single occurrence of the term “elderly” in the manuscript and ensured consistent use of inclusive alternatives such as “older adults” or “aging population.”

We are grateful for your careful review and believe that these revisions have strengthened the manuscript. Thank you again for your time and expertise.

Sincerely,
Rodrigo Cappato de Araújo (on behalf of all authors)

Reviewer 2 Report

Comments and Suggestions for Authors

Thank you for this exceptionally well-written article on a very timely topic.  

This  FREVO (Fatores de Risco no Envelhecimento) study protocol presents a robust, six-year longitudinal research protocol designed to investigate the interplay between intrinsic capacity and a broad range of risk factors among community-dwelling older adults in Petrolina, Brazil. Rooted in the WHO’s ICOPE (Integrated Care for Older People) framework, this study aims to generate localized, evidence-based insights into aging in a low-resource setting—particularly in Brazil’s semi-arid northeast.

Strengths

Commendably, the FREVO study is aligned with the WHO's intrinsic capacity model, emphasizing functional ability, independence, and healthy aging. By operationalizing the ICOPE framework in a Brazilian context, the study provides a valuable opportunity to assess the adaptability of global concepts in resource-limited environments.

A key strength of the study is its multidomain approach. The research protocol integrates cognitive, psychological, sensory, mobility, vitality, and environmental dimensions. Furthermore, it includes risk factors often underexamined in aging research—such as urinary incontinence, sarcopenia, sleep, pain, and life space mobility.

The study design is methodologically sound. The use of validated tools adapted for the Brazilian population ensures reliability and cultural relevance. Annual in-person assessments and biannual follow-up calls enable a balanced combination of depth and feasibility. The anticipated sample size (n=496), stratified by sex and age, provides a representative cross-section of the local older adult population.

The planned use of Latent Profile Analysis (LPA) to identify risk phenotypes and Cox regression models for time-to-event outcomes reflects a sophisticated and contemporary analytical strategy. This approach supports person-centered interpretations and strengthens the study’s predictive power for adverse aging outcomes.

By identifying modifiable risk factors and high-risk phenotypes in a socioeconomically vulnerable population, the FREVO study has the potential to inform primary care interventions, public health policy, and localized strategies for directly promoting functional aging.

Limitations you might want to consider.

I wonder if your predicted 15% loss is overly optimistic. Given the six-year follow-up period, participant attrition is an expected challenge. Although the authors account for a 15% loss to follow-up in their sample size calculation, maintaining engagement and data completeness over time—especially via phone follow-up—may require adaptive strategies.

The exclusion of older adults with serious cardiovascular, cognitive, or neuromuscular conditions, while methodologically justifiable, could omit high-risk subgroups who may benefit significantly from targeted interventions. Sensitivity analyses or parallel studies including these populations might be warranted.

While the study admirably incorporates external factors,  environmental and social factors, there may be limitations in capturing dynamic and systemic elements (e.g., healthcare system changes, climate impacts, political instability) that affect aging trajectories in low-resource settings over time.

Contributions

The FREVO study represents an important contribution to global aging research. It addresses the urgent need for contextually relevant data from the Global South, particularly from regions facing compounded social, economic, and environmental challenges. The study is poised to enhance our understanding of how intrinsic capacity interacts with modifiable risk factors to shape health trajectories in later life.

Readability.

This is a very readable, well written article.  It  follows a logical and well-organized structure typical of health research protocols—starting with the background and rationale, moving through objectives, methods, instruments, and anticipated outcomes. The tone is professional, and the technical terminology is appropriate for the academic audience of this journal.

The section describing measurement tools could benefit from tighter editing. You could simplify some of the highly detailed descriptions, as they may overwhelm some readers.

There is a readability leap between theoretical frameworks and operational tools. It might be useful to add a clear bridging sentence to improve the readability.  

The readability suggestions are minor! Overall, the FREVO protocol is a timely, well-conceived, and methodologically rigorous study that stands to generate meaningful insights into aging in underrepresented contexts. Its application of global frameworks to local realities, use of validated multidimensional assessments, and advanced analytic strategies offer a valuable model for future aging research in similar settings. If successfully implemented and sustained, the findings could lead to tailored health interventions and inform national aging policies in Brazil and the rest of the world. I will wait to see the report in 7 or 8 years. 

Author Response

Dear Reviewer,

Thank you very much for your generous and thoughtful review of our manuscript. We are truly honored by your encouraging remarks and your recognition of the relevance, methodological rigor, and potential impact of the FREVO study. Your detailed and constructive feedback has been extremely valuable, and we are grateful for the opportunity to improve our manuscript based on your observations.

We address your comments as follows:

Loss to follow-up over the six-year period
We fully agree that maintaining engagement in a longitudinal study of this duration can be challenging. Although we estimated a 15% loss to follow-up based on previous studies in similar contexts, we acknowledge this may be optimistic. We have now revised the Statistical Analysis section to explicitly describe our planned strategies to mitigate attrition, including regular contact with participants’ family members or caregivers, small engagement incentives, and the involvement of community health agents for retention support. Sensitivity analyses will also be conducted to explore the impact of attrition on study outcomes.

Exclusion of individuals with severe health conditions
Your point about the exclusion of older adults with severe cardiovascular, neuromuscular, or cognitive conditions is well taken. While this decision was made to ensure the feasibility of follow-up and data integrity, we recognize that these individuals represent an important and vulnerable subgroup. We have now included a note in the Considerations section acknowledging this limitation and suggesting that future studies—or secondary analyses from the FREVO cohort—should explore specific strategies to include these high-risk populations.

Broader systemic and environmental influences
We agree that structural and systemic factors—such as healthcare reforms, climate variability, and political context—may influence aging trajectories over time. Although we aim to capture some of these aspects indirectly through socioeconomic and geographic variables, we have included a note in the Considerations section acknowledging the limitations of our capacity to measure more dynamic systemic influences. We plan to incorporate contextual indicators into future waves or linked analyses when feasible.

Readability and integration of theory and tools

Thank you for your helpful suggestions regarding readability. We made minor adjustments to improve clarity and flow, particularly by adding a bridging sentence between the theoretical framework and the measurement instruments.

However, as this is a protocol article, we intentionally chose to provide a more detailed description of the methods and assessment tools to ensure transparency and reproducibility of the research process. In future publications arising from this study, we plan to present more concise descriptions of the instruments, focusing on their application and results.

Once again, we deeply appreciate your generous feedback, which reinforces the importance of our research and provides meaningful suggestions to strengthen it further. We are excited about the long-term potential of the FREVO study to contribute to aging science and inform public health strategies in Brazil and beyond.

Warm regards,
Rodrigo Cappato de Araújo (on behalf of all authors)

Reviewer 3 Report

Comments and Suggestions for Authors

Dear Authors,

After reading your manuscript, I have some questions/issues that require the attention listed below:

  1. Please bear in mind that terms like “elder” or “elderly” evoke negative stereotypes of older adults, which can lead to othering older adults, bias against older adults, and poor outcomes for older adults. Instead of those terms, more neutral phrases are preferred, such as “older adult, “older person,” or “persons over 65.” I suspect that you are fully aware of it because you mainly use other terms, and it happened only by accident (only once in the whole paper);
  2. I am impressed with the rigorous protocol and the many validated tools chosen to assess the prevalence of intrinsic capacity decline among older adults. I hope that your study will successfully investigate the associated risk factors, contributing to a better understanding of the specific challenges faced by a senile population.

Best regards,

The reviewer.

Author Response

Dear Reviewer,

Thank you very much for your kind words and for recognizing the methodological rigor of our study protocol, as well as the relevance of the tools selected to assess intrinsic capacity among older adults. We are grateful for your support and encouragement.

Regarding your important comment on the use of terminology, we fully agree that terms such as "elder" or "elderly" can perpetuate stereotypes and should be avoided in scientific communication. As you correctly noted, our manuscript predominantly uses the term "older adults." The isolated occurrence of "elderly" was indeed unintentional, and we have now revised the manuscript to ensure consistent and respectful language throughout.

We appreciate your attentive reading and thoughtful feedback, which contributes to the improvement of our work in both scientific and ethical dimensions.

Warm regards,
Rodrigo Cappato de Araújo (on behalf of all authors)

Reviewer 4 Report

Comments and Suggestions for Authors
  • The study is based solely on 496 older adults from the Petrolina region of Brazil, which may limit its external validity. It is recommended that the "Discussion" section explicitly specifythe geographical limitations of the sample and suggest that future research include more diverse regions (such as rural areas or cities with different levels of development) to validate the generalizability of the findings.
  • This study spanned six years, which may have led to loss to follow-up. It is recommended to provide a detailed explanation of how missing data were handled and to report the actual loss-to-follow-up rate and its potential impact in the "Results" section.
  • Some of the cited references are outdated. It is recommended to update them with studies from the past five years to enhance the timeliness and relevance of the research.

Author Response

Dear Reviewer,

We sincerely thank you for your valuable feedback on our manuscript. Your comments have helped us to improve the clarity and scientific rigor of the study protocol. Below, we address each of your suggestions:

External Validity and Geographical Limitations
We agree with your observation regarding the limited geographical scope of our sample. In response, we have included a statement in the Considerations section acknowledging this limitation. We now explicitly mention that the sample is restricted to older adults living in an urban area of the Petrolina region and that future studies should consider including participants from rural areas and cities with varying levels of development to enhance the external validity and generalizability of the findings.

Loss to Follow-Up and Missing Data
As this is a protocol paper, we have not yet reported results. However, we acknowledge the importance of clarifying how we plan to handle missing data. Therefore, we have added a paragraph in the Statistical Analysis section detailing the strategies for managing missing data, including the use of multiple imputation and sensitivity analyses, as well as plans to report the loss-to-follow-up rate once data collection is complete.

Outdated References

We appreciate your observation regarding the age of some references. In our manuscript, many of the cited works refer to the validation studies of instruments and scales used in the assessment of intrinsic capacity and related domains. These references were selected intentionally to support the validity and cultural adaptation of these tools to the Brazilian context, which remains highly relevant.
Additionally, some of the references cited—such as those related to the WHO’s ICOPE framework—are seminal works that represent the original formulation of key concepts underlying the study rationale. Whenever possible, we have complemented these with more recent publications to enhance the timeliness of the literature base and ensure that our citations reflect both foundational and up-to-date contributions.

Reviewer 5 Report

Comments and Suggestions for Authors

Thank you for the opportunity to review this protocol paper on the FREVO study. The article clearly introduces the research design and data collection process, demonstrating strong scientific merit and practicality. Here are my brief comments:

Strengths

  • Clear Design: The study has clear objectives, detailed methods, and a rational data collection process, making it easy to understand and implement.
  • Practical Integration: Applying the WHO’s intrinsic capacity framework to the low-resource context in Brazil is innovative and relevant.

Suggestions

  • Enhance Background: Provide more specific data on the aging situation in Brazil.
  • Discuss Limitations: Briefly mention potential limitations of the study and directions for future improvements.
  • Highlight Significance: Further emphasize the potential impact of the study results on policy and practice.

This is a high-quality protocol paper with a well-structured research design, clear objectives, and scientific data collection and analysis methods. I recommend acceptance with minor revisions based on the above suggestions. I look forward to the success of the research team.

Author Response

Dear Reviewer,

Thank you very much for your thoughtful and encouraging review of our manuscript. We are grateful for your recognition of the scientific merit, clarity of design, and practical relevance of the FREVO study protocol. We especially appreciate your constructive suggestions, which have contributed to improving the manuscript.

In response to your comments:

Enhance Background – Provide more specific data on aging in Brazil
We have revised the Introduction to include more detailed and updated data regarding the aging population in Brazil, including regional disparities and projections for the coming decades. These additions help frame the urgency and contextual relevance of the study.

Discuss Limitations – Briefly mention potential limitations
We have included a short paragraph in the Considerations section addressing potential limitations such as attrition bias, self-reported data, and possible underrepresentation of older adults with severe impairments.

Highlight Significance – Emphasize potential impact on policy and practice
The final paragraph of the Considerations section has been revised to reinforce the potential contributions of the FREVO study to public health policy, especially in underserved regions of Brazil. We also emphasize how the findings may inform the implementation of WHO's healthy aging strategies in similar low-resource settings.

Thank you once again for your insightful feedback and support for our work.

Sincerely,
Rodrigo Cappato de Araújo (on behalf of all authors)

Round 2

Reviewer 4 Report

Comments and Suggestions for Authors
  • The manuscript clearly states that individuals with severe cardiovascular, neuromuscular, and cognitive impairments were excluded from the study. While this approach helps ensure data quality, it also limits the applicability of the findings to high-risk populations. It is recommended that the authors propose concrete strategies for including these groups in future research.
  • Some lengthy sentences reduce readability,it is recommended that the manuscript undergo thorough language editing to ensure clarity and conciseness throughout the text.

Author Response

The manuscript clearly states that individuals with severe cardiovascular, neuromuscular, and cognitive impairments were excluded from the study. While this approach helps ensure data quality, it also limits the applicability of the findings to high-risk populations. It is recommended that the authors propose concrete strategies for including these groups in future research.

We thank the reviewer for this important observation. We agree that including high-risk populations in future research is crucial for expanding the understanding of aging trajectories. However, the current study was designed to investigate intrinsic capacity and its associated factors in community-dwelling older adults capable of responding to the full battery of assessments. The exclusion criteria were adopted to ensure methodological rigor, uniform application of instruments, and internal validity of the findings.

While we acknowledge the importance of advancing future research that incorporates more vulnerable populations, we believe that proposing concrete strategies for including individuals with severe impairments falls outside the scope of the present study. Such inclusion would require extensive adjustments to recruitment procedures, ethical safeguards, measurement tools, and data collection logistics, elements that deserve dedicated methodological planning beyond what can be addressed in the context of this manuscript.

Nevertheless, we have acknowledged this limitation in the revised version of the manuscript and emphasized the need for complementary studies that specifically address these high-risk groups.

  • Some lengthy sentences reduce readability,it is recommended that the manuscript undergo thorough language editing to ensure clarity and conciseness throughout the text.

We appreciate the reviewer’s observation. In response, we have thoroughly revised the manuscript to improve clarity and readability. Several sentences and paragraphs were restructured to reduce length and enhance textual fluency. Additionally, the revised version was reviewed by a native English-speaking co-author